# A Comprehensive Review of the Usefulness of Prebiotics, Probiotics, and Postbiotics in the Diagnosis and Treatment of Small Intestine Bacterial Overgrowth

**DOI:** 10.3390/microorganisms13010057

**Published:** 2025-01-01

**Authors:** Adrian Martyniak, Magdalena Wójcicka, Iwona Rogatko, Tomasz Piskorz, Przemysław J. Tomasik

**Affiliations:** 1Department of Clinical Biochemistry, Pediatric Institute, Faculty of Medicine, Jagiellonian University Medical College, 30-663 Krakow, Poland; adrian.martyniak@uj.edu.pl (A.M.); magdalena.wojcicka@uj.edu.pl (M.W.); iwona.rogatko@uj.edu.pl (I.R.); 2Chair in Gynecology and Obstetrics, Faculty of Medicine, Jagiellonian University Medical College, 31-008 Krakow, Poland; t.piskorz@uj.edu.pl

**Keywords:** microbiota, intestinal, gut, antibiotics, breath test, tryptophan, inulin, butyrate, fatty acids, bacteria

## Abstract

Small intestinal bacterial overgrowth (SIBO) is a disorder characterized by the excessive growth of bacteria in the small intestine. Bacterial overgrowth disrupts the bacterial balance and can lead to abdominal pain, weight loss, and gastrointestinal symptoms, including bloating, diarrhea, and malabsorption. SIBO is widespread in the population. There are two main methods for diagnosing SIBO: breath tests and bacterial culture. The most commonly used method is a breath test, which enables the division of SIBO into the following three types: hydrogen-dominant (H-SIBO), methane-dominant (CH_4_-SIBO), and hydrogen/methane-dominant (H/CH_4_-SIBO). This comprehensive review aims to present the current knowledge on the use of prebiotics, probiotics, and postbiotics in the context of SIBO. For this purpose, medical databases such as MEDLINE (PubMed) and Scopus were analyzed using specific keywords and their combinations. This review is based on research studies no older than 10 years old and those using only human models. In summary, clinical studies have shown that the efficacy of SIBO therapy can be increased by combining antibiotics with probiotics, especially in vulnerable patients such as children and pregnant women. The further development of diagnostic methods, such as point of care testing (POCT) and portable devices, and a better understanding of the mechanisms of biotics action are needed to treat SIBO more effectively and improve the quality of life of patients.

## 1. Introduction

Small intestinal bacterial overgrowth (SIBO) is a disorder characterized by the excessive growth of bacteria in the small intestine, defined as a bacterial population in the small intestine exceeding 10^5^ colony-forming units (CFU) per ml of jejunal juice [1,2]. Bacterial overgrowth disrupts the bacterial balance and can lead to abdominal pain, weight loss, and gastrointestinal symptoms, including bloating, diarrhea, and malabsorption of nutrients, which can even lead to osteoporosis. The exact prevalence of SIBO in the general population is unknown. Depending on the study, the reported incidence is between 2.5 and 22% [2]. In patients with gastroenterological disorders, SIBO has been detected in 33.8% of cases [3]. This prevalence increases with age and in populations with comorbidities. The risk factors of SIBO are: delayed orocecal transit time [OCTT], reduced hydrochloric acid secretion, and reflux from the colon to the small intestine due to ileocecal valve dysfunction [2,4,5]. Smoking and anemia strongly increased the risk of SIBO (odds ratio [OR] = 6.66 and OR = 4.08, respectively) [6]. Age increased the risk of SIBO (OR = 1.04) [7]. SIBO prevalence did not depend on gender or race [8]. Research showed that diet and drugs also can influence SIBO prevalence. In preschool children on a long-term dairy-free diet, SIBO was detected more frequently than in children on a typical diet (55% and 20%, respectively) [9]. A very serious problem may be using proton pump inhibitors (PPI); the risk of SIBO is escalated for those on this medication (OR = 1.71) [10].

There are two main methods for diagnosing SIBO: breath tests and bacterial culture, and both of them have limitations [11]. The breath test is simple to perform, comfortable for the patient, and low-cost. But, unfortunately, the interpretation of the results can be difficult. Breath tests are also used for the diagnosis of carbohydrates intolerance, and these diseases might interfere with SIBO diagnosis. The most widely used breath tests are the glucose and lactulose breath tests. Both tests are based on measured hydrogen and/or methane in exhaled air. In 2017, the North American consensus concluded that “a rise in hydrogen of ≥20 ppm above the baseline value by 90 min following substrate ingestion during glucose or lactulose breath test for SIBO was considered positive” [12]. These components in exhaled air enable the division of SIBO into the following three types: hydrogen-dominant (H-SIBO), methane-dominant (CH_4_-SIBO), and hydrogen/methane-dominant (H/CH_4_-SIBO) [13].

The direct method of diagnosis for SIBO is counting bacterial colonies from small bowel luminal content cultures [11]. A bacterial count from the proximal jejunum of above 10^5^ CFU/mL has been traditionally accepted as positive for SIBO [14]. This method also has its drawbacks. The most important ones include the following: (1) small bowel intubation; (2) many bacterial species do not grow in routine media and/or under culture conditions (aerobic/anaerobic); and (3) there is a high risk of contamination. For these reasons, this ‘gold standard’ is currently rarely used in SIBO diagnosis. A different approach for the diagnosis of SIBO was undertaken by Ardatskaia et al. They proved that short-chain fatty acids (SCFA) evaluation in biological materials, such as fecal and duodenal content, may be a better SIBO indicator than breath tests [15]. SCFAs can also be used to determine the intensity of SIBO using high-resolution ^1^H nuclear magnetic resonance (NMR) spectroscopy in upper gut aspirate. Bala et al. showed that, in patients with malabsorption syndrome (MAS) and SIBO, the quantity of acetate correlated positively with the total colony counts of bacteria [16]. Postbiotics can also be used to assess the type of SIBO. In a study by Wielgosz-Grochowska et al., serum folic acid (FA) correlated with methane (CH_4_-SIBO) in the breath test (r = 0.637, *p* = 0.002) [17]. This observation was confirmed by Platovsky et al. In their study, patients with increased FA levels had a 1.75 times greater likelihood of having SIBO than patients with typical FA levels [18]. This relationship arose because almost 45% of gut bacteria have the ability to synthesize FA. Similar dependence was observed in a study by Signoretti et al., where patients with chronic pancreatitis and SIBO had higher serum FA levels than patients without SIBO (16.5 ng/dL vs. 8.32 ng/dL; *p* = 0.05) [19]. It is considered secondary to the bacterial fermentation of substrates present in the intestinal lumen.

The management of SIBO is particularly challenging due to its complex etiology, multifactorial nature, and the tendency for recurrence after treatment. SIBO treatment is varied and often depends on the underlying disease and concomitant digestive system disorders. The goals of treatment for SIBO are threefold, as follows: (1) the correction of the underlying cause; (2) proper nutritional support; and (3) a reduction in bacteria using antibiotics. Therefore, there are possible surgical, dietary, and pharmaceutical therapies. Surgery may be beneficial in patients with small bowel diverticulosis, fistulas, or strictures which enhance the risk of SIBO and worsen its course. Dietary treatment is important for malnourished patients and/or those with deficiencies, e.g., vitamins; however, a healthy, balanced diet is recommended for all patients. Antibiotics are the basis of SIBO treatment; those frequently used in treatment include rifaximin, metronidazole, neomycin, norfloxacin, amoxicillin, and tetracycline. However, using antibacterial treatment sometimes generates complications, e.g., microbial resistance, drug interactions and side effects, dysbiosis with its severe possible symptoms, and, often, the need for repeat antibiotic treatments [20]. For these reasons, other methods of treating SIBO are being sought.

One promising area of research involves biotic agents—prebiotics, probiotics, and postbiotics. These factors are potentially able to modulate the gut microbiome in human organism-friendly ways that promote health and act against dysbiosis and overgrowth.

According to the expert consensus of the International Scientific Association for Probiotics and Prebiotics (ISAPP), a prebiotic is “a substrate that is selectively used by host microorganisms that confer a health benefit” [21]. Prebiotics must have the ability to resist digestion and absorption from the gastrointestinal tract but could be digested (fermented) by intestinal bacteria. The products of bacterial digestion lower the pH of the intestinal content and stimulate the growth of beneficial bacteria in the small bowels, such as the *Lactobacillus*, *Bifidobacterium*, and *Bacteroides* families. Finally, prebiotics have a positive effect on the health of the host [22]. The most common prebiotics are inulin, galactooligosaccharides (GOS), fructooligosaccharides (FOS), human milk oligosaccharides (HMO), xylooligosaccharides (XOS), mannanoligosaccharides (MOS), lactulose, and derivatives of galactose and β-glucans [23]. Polyphenols, polyunsaturated fatty acids (PUFA), and glucooligosaccharides are considered potential prebiotics [21]. Prebiotics are naturally present in food, in over 36,000 products. They are mostly of plant origin, e.g., wheat, asparagus, onions, chicory, and garlic, but can also be found in honey and cow’s milk. Moreover, prebiotics may also be added to food to increase its nutritional and health value.

Probiotics are defined as “live microorganisms which when administered in adequate amounts confer a health benefit on the host” [24]. These microorganisms may produce antimicrobial substances, modulate the host’s immune system response, protect from pathogenic bacteria adhesion to the epithelium, stimulate mucosal IgA production, and inhibit bacterial toxin production [25,26]. *Lactobacilli*, along with several species of *Bifidobacterium*, have historically been common probiotics. Nowadays, the list of probiotics is much longer. Naturally, fermented foods such as kefir, pickles, yogurt, and kimchi are rich in probiotics. Dietary supplements containing specific strains of probiotic bacteria have gained significant popularity [27]. The most important group of intestinal bacteria is lactic acid-producing bacteria (LAB), which produce lactic acid during the fermentation of saccharides. In addition, probiotics help in the production of short-chain fatty acids (SCFAs), vitamins, and bactericidins and play a role in the metabolism of bile acid salts [28].

Postbiotics are a relatively new concept and refer to bioactive compounds produced by microorganisms. Postbiotics, informally called metabiotics, are the structural components of probiotic microorganisms, their signaling molecules, and metabolites with a determined chemical structure. Also, postbiotics can optimize the host’s physiological functions and modulate metabolic and/or behavioral reactions related to the activity of the host’s indigenous microbiota [29]. According to a new definition (2021) prepared by ISAPP, postbiotics are “a preparation of inanimate microorganisms and/or their components that confers a health benefit on the host” [30]. Postbiotics are found in all natural fermented products. Postbiotics can be divided into a few groups, such as vitamins (e.g., folic acid and B12), organic acids including SCFAs (e.g., butyrate, propionate, and acetate), and amino acids (AAs) (e.g., tryptophan (Trp)). The positive effect of postbiotics can be direct or indirect. The direct effect depends on the action of the host cell, whereas the indirect effect is based on environmental changes in the gut [31]. The action mechanism of postbiotics is widespread [32]. The most important comprises immunomodulation by regulatory T cells (Tregs), mostly by SCFAs [33]. Postbiotics also have anti-inflammatory properties, e.g., blocking the synthesis of proinflammatory cytokines [34]. The antimicrobial activity of some peptides and bacteriocins may influence SIBO development [35]. Some amino acids also have a huge impact on the gut barrier [36].

This comprehensive review aims to present the current knowledge on the use of prebiotics, probiotics, and postbiotics in the context of SIBO, with a focus on their potential clinical applications and mechanisms of action in the management and treatment of this disease.

The database search used was exhaustive. MEDLINE (PubMed) and Scopus databases were used to analyze the available literature. Keywords and their combinations were used, such as probiotic, prebiotic, postbiotic, small intestinal bacterial overgrowth (SIBO), inulin, glucooligosaccharides (GOS), fructooligosaccharides (FOS), human milk oligosaccharides (HMO), short-chain fatty acid (SCFA), tryptophan, butyric acid, propionic acid, acetate acid, *Saccharomyces boulardii*, and lactic acid bacteria (LAB). The following inclusion criteria were used: research studies no older than 10 years old and studies on human models. The following exclusion criteria were used: animal studies models, studies older than 10 years, studies with incomplete or residual data.

The gray literature and additional sources are not included because we cannot guarantee a high quality of research.

## 2. Prebiotics and Probiotics in SIBO

The most widely used and studied probiotic strain for SIBO is *Saccharomyces boulardii*. These yeasts are a promising option in SIBO treatment because they are unaffected by antibiotics given to reduce gut bacterial flora, so can be co-administrated with antibiotics therapy [37,38]. Also, yeast strains reach the intestines in CFU numbers greater than those of bacterial probiotics, due to their resistance to the gastric environment. In addition, in the gut, yeast metabolites stimulate the growth of probiotic bacteria [39]. 

In a study by Redondo-Cuevas et al., the probiotic *Saccharomyces boulardii* was administered to 123 patients with SIBO (at a dose of 250 mg per day/about 5 bln CFU) along with antibiotic therapy, comprising 200 mg of rifaximin (two tablets, three times a day) and 500 mg of neomycin (one tablet twice a day). If symptoms persisted or side effects occurred, the dose was reduced to one tablet per day. Patients were also supplemented with essential oils, namely Oleocaps 2 (Pranarom), black cumin oil (Sura Vitasan), and wormwood (Nutri Holistic). Their compounds have anti-inflammatory and antimicrobial activity but not prebiotic properties. For 6 weeks after antibiotic therapy, *Bifidobacterium longum* supplementation, L-glutamine (in a dose of 5 g twice a day), and a low-FODMAP (fermentable oligosaccharides, disaccharides, monosaccharides, and polyols) diet was enforced. Such a scheme of combined therapy did not significantly change the breath test results. However, this combined treatment improved clinical outcomes and alleviated gastrointestinal symptoms, particularly in patients with SIBO associated with increased methane production [40].

Supplementation with *Saccharomyces boulardii* has also been effective in patients with SIBO and systemic sclerosis (SSc). In a study by García-Collinot et al., patients with SSc and SIBO were divided into three groups depending on their treatment regimen: 13 of them used metronidazole (500 mg twice daily for 7 days); 14 of them used *Saccharomyces boulardii* (200 mg twice a day); and 13 of them used metronidazole (500 mg) plus *Saccharomyces boulardii* (200 mg twice a day). All treatments were administered in the first week of a month or during the first and second week in two consecutive months, for the group that received both an antibiotic and a probiotic. *Saccharomyces*, whether used in combination with metronidazole or as a monotherapy, reduced SIBO, by 55% and 33%, respectively, compared to the sole metronidazole treatment (25%), and alleviated side effects, such as upper abdominal burning, bloating, and diarrhea. It was concluded that *Saccharomyces boulardii* mitigated the discomforts related to SIBO [41].

Efremova et al. also administered *Saccharomyces boulardii* twice a day at a dose of 250 mg, but for three months, to 20 patients with SIBO and cirrhosis. This treatment eliminated SIBO in 80% of patients. Additionally, a reduced incidence of ascites and hepatic encephalopathy was observed, with a reduced severity of cirrhosis, accompanied by a better prognosis for the patient [42]. Although some studies have shown probiotic efficacy in reducing the risk of developing hepatic encephalopathy and SIBO-related symptoms in patients with chronic liver disease, results regarding their effects on other health parameters, such as intestinal permeability or indices of liver function, remain inconclusive [43,44]. As demonstrated by Lunia et al., probiotics not only reduced SIBO but may have also prevented the occurrence of hepatic encephalopathy (HE) in patients with cirrhosis. Patients with cirrhosis without a previous history of hepatic encephalopathy received probiotics containing the following: *Bifidobacterium breve*, *Bifidobacterium longum*, *Bifidobacterium infantis*, *Lactobacillus acidophilus*, *Lactobacillus plantarum*, *Lactobacillus paracasei*, *Lactobacillus bulgaricus*, and *Streptococcus thermophilus* (110 million CFU) as one capsule, taken three times a day for 3 months. Patients in the study group had a lower incidence of hepatic encephalopathy than those in the control group. The supplemented group also showed a reduction in SIBO, orocecal transit time, and arterial ammonia, as well as an improvement in the psychometric hepatic encephalopathy score, critical flicker frequency, and reversal of minimal HE [45]. However, a study by Kwak et al., conducted on patients with chronic liver disease, showed that 4 weeks of probiotic therapy (*Bifidobacterium bifidum*, *Bifidobacterium lactis*, *Bifidobacterium longum*, *Lactobacillus acidophilus*, *Lactobacillus rhamnosus*, and *Streptococcus thermophilus;* 5 × 10^9^ of living cells), alleviated SIBO and gastrointestinal symptoms, but did not affect intestinal permeability or contribute to an improvement in liver parameters or Child–Pugh scores (a clinical tool used to assess the severity of chronic liver disease, which evaluates the following five key factors: serum bilirubin, serum albumin, prothrombin time (INR), ascites, and hepatic encephalopathy, assigning each a score from one to three) [46,47].

SIBO is also a frequent problem in obese patients after Roux-en-Y Gastric Bypass (RYGB). This is one of the most effective surgical treatments for obesity but can lead to complications such as the development of SIBO. Consequently, the role of probiotics as a post-surgical supportive medication has been investigated. Despite some promising results regarding reductions in symptoms like bloating, the evidence for the effectiveness of probiotics in preventing or eliminating SIBO after RYGB remains limited [48]. Wanger et al. showed that patients one year after the procedure who used probiotics consisting of 50 million CFUs per capsule (*Lactobacillus acidophilus*, *Bifidobacterium lactis*, *Lactobacillus rhamnosus*, *Bifidobacterium longum*, *Lactobacillus plantarum*, *Bifidobacterium bifidum*, and *Lactobacillus gasseri*) for 8 weeks did not show a significant clinical improvement or reduction in SIBO compared to a placebo. Thus, they suggested considering the use of other strains in probiotic preparations or other methods of alleviating the SIBO symptoms in RYGB patients [49]. Wagner et al., in another study, showed that supplementing patients with one daily dose of probiotics lasting from 7 days after the RYGB procedure for the next 90 days (5 billion *Lactobacillus acidophilus* and 5 billion *Bifidobacterium lactis*) also had no effect on the development of SIBO, but was associated only with a reduction in flatulence [50].

Probiotics have also been found to be useful in reducing disturbances resulting from SIBO in a pediatric population. In children, SIBO symptoms do not differ from those noted in adult patients [5].

In a study by Peinado Fabregat et al., the addition of a probiotic to the antibiotic treatment in pediatric patients with SIBO increased the number of children with partial or complete resolution of symptoms. In patients taking the probiotic *Lactobacillus rhamnosis* alone or in combination with an antibiotic, symptoms resolved in 81.2% of patients, while in patients treated with antibiotics alone, they resolved in 67.7% of patients [51]. Ockeloen et al. showed the effectiveness of daily probiotic supplementation for 8 weeks in children with SIBO at age-appropriate doses: aged 1–4 years with 2 g of powder 1 × 10^9^ CFU/g *Bifidobacterium* and *Lactobacillus* for children aged 1–4 years, and one capsule 1 × 10^9^ CFU/g *Bifidobacterium* and *Lactobacillus* for children aged 5–18 years. They found reduced severity of abdominal complaints related to SIBO after 5 months of therapy in 70% of children and in 40% after 15 months. The authors suggested a repeat cycle of probiotic administration to maintain the desired effect [52]. Table 1 shows a comparison of the results of probiotic therapy between different groups of patients, including children with SIBO, as well as the limitations of each study.

Bacterial overgrowth is common in pregnant women with hypothyroidism (50–60% vs. 25–30% of pregnant women with typical thyroid function). The gut microbiota may influence the absorption of orally administered hormones, thus influencing thyrotropin (TSH) [53]. In pregnant hypothyroidic women, the use of probiotics to eliminate SIBO and improve the effect of hormone supplementation seems to be safer for the fetus and mother, compared to antibiotics.

Ouyang et al. treated 74 typical pregnant women and 78 pregnant women with subclinical hypothyroidism. Both study groups received probiotics, as follows: *Bifidobacterium infantis* 2.7 × 10^8^ CFU/g, *Lactobacillus acidophilus* 4.7 × 10^8^ CFU/g, *Enterococcus faecalis* 6.1 × 10^7^ CFU/g, *Bacillus cereus* 1.5 × 10^6^ CFU/g at a dosage of 1.5 g, three times a day, in combination with prebiotics in the form of a dietary supplement containing inulin, ice sugar, microcrystalline cellulose, and oat fiber at a dose of 5 g, three times a day, for 21 days. That therapy reduced bacterial overgrowth (based on the results of the lactulose methane/hydrogen breath test), mainly CH_4_-SIBO, by 63.6%. However, probiotic supplementation had a positive effect, resulting in better absorption of levothyroxine, a decrease in TSH, and the normalization of thyroid function in all intervention groups [54]. In a study by Zhang et al., probiotic treatment with *Bifidobacterium infantis* (≥0.5 × 10^6^ CFU/g), *Lactobacillus acidophilus* (≥0.5 × 10^6^ CFU/g), *Enterococcus faecalis* (≥0.5 × 10^6^ CFU/g), and *Bacillus cereus* (≥0.5 × 10^5^ CFU/g) at a dosage of 1.5 g, three times a day for 21 days, was conducted for pregnant women with SIBO. SIBO conversion rates in pregnant women with hypothyroidism, and in pregnant women without thyroid disorders who were treated with probiotics, were 71.4% and 64.3%, respectively. This treatment also reduced clinical SIBO symptoms, improved thyroid function, and reduced the levels of inflammatory markers [55]. Hao et al., in a similar study on hypothyroidic pregnant women, found that the overall SIBO recovery rate for SIBO in pregnant women using probiotics was 53.6% and, in the CH_4_-SIBO group, the negative conversion rate was 90.7%. The probiotic treatment influenced not only drug absorption but also had an impact on the body’s inflammatory response and the integrity of the intestinal membrane [56].

A comparison of the results of treating SIBO with probiotics in pregnant women with hypothyroidism is provided in Table 2, where different probiotic strains, dosages, and SIBO negative conversion rates are summarized and the limitations of each study are also presented.

## 3. Postbiotics in SIBO

The intestinal microbiota can impact many systems in the human body, and these effects are more profound in SIBO patients. Changes in SCFAs indicate a violation of the gut microbiocenosis in patients with allergic bronchial asthma (BA). A reduction in SCFA levels is one of the first signals of reduced gut *Bifidobacterium* and *Lactobacterium* activity. SIBO in this group of patients may be considered as an impairing factor [57]. In 2022, Ozimek et al. checked the impact of SCFAs in patients with BA and SIBO. In patients with BA, regardless of whether they were allergic or non-allergic, there was a decrease in SCFA. Thirty patients on standard BA therapy were divided into the following three groups: 10 patients with SIBO were prescribed rifaximinium (200 mg three times per day) for a week; 10 patients with SIBO were prescribed rifaximinium (200 mg three times per day) for a week and LAB probiotics (one capsule, three times per day—3.0 × 10^9^ CFU/capsule at least) for a month; and 10 patients without SIBO were also administered LAB probiotics (one capsule, three times per day—3.0 × 10^9^ CFU/capsule at least) for a month. After the treatment, all patients noted the normalization of the fecal SCFA spectrum and anaerobic index (AI) (AI = (butyric acid + propionic acid)/acetic acid). Non-SIBO patients had significantly higher levels of SCFA after treatment (*p* < 0.001). Patients with SIBO after antibiotic therapy and probiotic administration had a more favorable isoacids/acids ratio than patients only on antibiotics therapy (*p* < 0.05), similar to non-SIBO patients. Those results may indicate a positive effect of probiotic administration due to the normalization of the isoacids/acids ratio and the beneficial effects of postbiotics such as SCFAs in BA, by stabilizing the intestinal redox potential and pH (reduction in CO_2_ and H_2_ production, due to reduced bacterial overgrowth) [58].

Tryptophan (Trp) metabolism in the gut plays a key role in regulating the nervous and immune systems, and its disruption can lead to serious health problems. The main Trp catabolism pathway in the body is the kynurenine pathway, and its abnormal activation is associated with inflammation, cancer development, and neurodegenerative and psychiatric diseases. In a study by Chojnacki et al., patients with SIBO before antibiotic treatment had higher urinary kynurenine (KYN) and quinolinic acid (QA) levels compared to the control group, while their kynurenic acid (KYNA) levels were lower. After rifaximin treatment (daily dose of 1200 mg for 14 days, then 1200 mg for 10 days in the following two months), decreases in urinary levels of all metabolites were observed, suggesting that the treatment may have a beneficial effect on the regulation of the kynurenine pathway in patients with SIBO. SIBO-related changes to Trp metabolism on the KYN pathway also promote abdominal and mood disorders [59].

Trp metabolism is also involved in mucus production. In SIBO, an increase in Trp metabolites stimulates IDO-1 in epithelial cells and promotes the differentiation of secretory cells. The consequence of these changes is chronic diarrhea. In another study by Chojnacki et al., in patients with SIBO and chronic diarrhea (SIBO-D) and with chronic constipation (SIBO-C), Trp and its metabolites, such as KYN, QA, 5-hydroxyindoleacetic acid (5-HIAA), and xanthurenic acid (XA), were measured in urine, using liquid chromatography with tandem mass spectrometry (LC-MS/MS), before and after antibiotic therapy. All SIBO patients were diagnosed with mild to moderate anxiety and mild depression. Patients with SIBO-C showed elevated levels of KYN, XA, and QA before treatment. After rifaximin treatment (10 days in doses of 1200 mg), significant decreases in 5-HIAA/TRP and KYN/TRP ratios were observed in the SIBO-D group, and decreases in KYN and QA levels were observed in the SIBO-C group. Anxiety and depression levels decreased in both groups. This suggests that therapy for SIBO has a beneficial effect on the local (gut) and systemic (psyche) status of patients via the modulation of tryptophan metabolism [60]. In another study, Chojnacki et al. analyzed the conversion of tryptophan to serotonin (Trp-5-HT pathway) in patients with SIBO. Patients with SIBO and chronic diarrhea (SIBO-D) showed reduced tryptophan hydroxylase type 1 (TPH-1) enzyme activity in the small intestinal mucosa and lower blood serotonin levels compared to controls and SIBO patients with chronic constipation (SIBO-C) (*p* < 0.001). In addition, urinary levels of 5-HIAA, the main metabolite of serotonin, were higher in SIBO-D patients than in SIBO-C patients and controls (*p* < 0.001 in both cases). After treatment with the antibiotic rifaximin, 5-HIAA levels were significantly reduced in both SIBO groups (*p* < 0.001). However, no changes in TPH-1 enzyme activity were described after treatment. These results suggest that reduced TPH-1 activity and altered serotonin levels may be important laboratory markers used to differentiate between SIBO complicated with diarrhea and SIBO complicated with constipation [61]. 

The authors of the study did not administer Trp because this would be considered a medical experiment. The authors examined the impact of SIBO and treatment on Trp endogenous metabolism.

## 4. Limitations of the Study

The major limitation of this review is the relatively small sample sizes in some studies. In addition, patients were often given combination drugs, which can bias the results. Conclusions are limited due to a lack of available or reliable data. These research results require confirmation in a multicenter well-designed study. For more details regarding the limitations in the above-described studies, please see Table 1 and Table 2.

## 5. Conclusions

The diagnosis and treatment of SIBO are challenging due to its complex etiology, tendency to recur, and its diverse symptoms such as bloating, diarrhea, abdominal pain, and nutrient malabsorption. SIBO is treated as a non-severe condition, but recovery might significantly improve health outcomes, diminish disturbances from the gut, and improve mood. Clinical studies have shown that the efficacy of therapy can be increased by combining antibiotics with probiotics. An example is the use of the *Saccharomyces boulardii* strain, which is characterized by antibiotic resistance and better gastrointestinal penetration. In the context of SIBO diagnosis, standard methods such as breath tests and bacterial cultures have their limitations. More accurate, objective assessment methods are needed to provide an accurate diagnosis and evaluate treatment progress. The use of breath tests to assess hydrogen and methane levels can help distinguish between types of SIBO, such as SIBO dominated by hydrogen or methane production. POCT, in the diagnosis of SIBO, provides a quick and simple assessment of patients, allowing for the monitoring of the exhaled gas profile. Portable devices give patients the opportunity to self-monitor treatment effects. POCT can support the dynamic adjustment of therapies, which can improve the effectiveness of SIBO treatment [62]. Further research is needed on precise diagnostic methods that could more objectively assess the conditions of SIBO patients, as well as monitor metabolic changes, such as tryptophan metabolism. It has been shown that changes in amino acid metabolism, particularly tryptophan metabolism, can help differentiate between forms of SIBO with diarrhea (SIBO-D) and constipation (SIBO-C).

In conclusion, the further development of diagnostic methods, such as POCT, and a better understanding of the mechanisms of action of prebiotics, probiotics, and postbiotics are needed to treat SIBO more effectively and improve the quality of life of patients.

## Figures and Tables

**Table 1 microorganisms-13-00057-t001:** Comparison of the use and efficacy of probiotics in different groups of adult and pediatric patients (lower part of the table) with SIBO.

Study	Number and Characteristics of Patients Who Underwent Probiotics Therapy	Dose and Schedule	Outcomes	Limitations of the Study
Redondo-Cuevas et al., 2024 [40]	123 patients with CH_4_-SIBO.	*Saccharomyces boulardii* (250 mg per day) for 10 days of antibiotic therapy (rifaximin 200 mg, two tablets three times a day; neomycin 500 mg, one tablet two times a day; in case of symptoms, dose was reduced to once a day).	No significant change in breath test results, but clinical scores and severity of gastrointestinal symptoms decreased.	No gold standard in unambiguous classification of breath test data and clinical symptoms.
García-Collinot et al., 2020 [41]	20 SSc patients with SIBO.	*Saccharomyces boulardii* (200 mg 2× a day for 7 days) vs. metronidazole (500 mg two times a day for 7 days) vs. *Saccharomyces boulardii* (200 mg two times a day) + metronidazole (500 mg two times a day) (7 days + 7 days *Saccharomyces boulardii*) for over 2 months on the first (and second) week of the month.	SIBO reduction: 55% with probiotic and metronidazole, 33% with probiotic, 25% with metronidazole alone; symptom reduction was also observed.	Small sample size, short follow-up period, <80% success rate in adherence to therapy.
Efremova et al., 2024 [42]	20 patients with cirrhosis and SIBO.	*Saccharomyces boulardii* (250 mg two times a day for 3 months).	Eradication of SIBO in 80% of patients with probiotic, compared to 23.1% in placebo group; decrease in incidence of ascites and hepatic encephalopathy.	Small sample sizes resulted in lack of direct comparison between eradicated and persistent SIBO in individual groups.
Lunia et al., 2014 [45]	86 patients with cirrhosis, 33 of whom also had SIBO.	*Bifidobacterium breve*, *Bifidobacterium longum*, *Bifidobacterium infantis*, *Lactobacillus acidophilus*, *Lactobacillus plantarum*, *Lactobacillus paracasei*, *Lactobacillus bulgaricus*, and *Streptococcus thermophilus* (110 million CFU), one capsule three times a day for 3 months.	Reductions in SIBO, orocecal transit time, and arterial ammonia and improvements in the psychometric HE scores, critical flicker frequency, and reversal of minimal HE; also, supplemented group had lower incident of HE than control.	Possible bias due to lack of blinding of given treatment.
Kwak et al., 2014 [46]	53 patients with chronic liver disease, 13 of whom also had SIBO.	*Bifidobacterium bifidum*, *Bifidobacterium lactis*, *Bifidobacterium longum*, *Lactobacillus acidophilus*, *Lactobacillus rhamnosus*, and *Streptococcus thermophilus*, with a dosage of 5 × 10⁹ living cells per capsule for 4 weeks.	Alleviated SIBO and gastrointestinal symptoms, no improvement in intestinal permeability, liver parameters, or Child–Pugh scores.	Small sample size.
Wagner et al., 2024 [49]	47 patients one year after Roux-en-Y gastric bypass (RYGB).	50 million CFUs per capsule of *Lactobacillus acidophilus*, *Bifidobacterium lactis*, *Lactobacillus rhamnosis*, *Bifidobacterium longum*, *Lactobacillus plantarum*, *Bifidobacterium bifidum*, and *Lactobacillus gasseri* for 8 weeks.	No significant clinical improvement or reduction in SIBO compared to placebo.	Lower than recommended fiber intake could influence results of study.
Wagner et al., 2021 [50]	73 patients post-RYGB.	One daily dose of probiotics: 5 billion *Lactobacillus acidophilus* and 5 billion *Bifidobacterium lactis* starting 7 days after surgery for 90 days.	No effect on SIBO development but associated with reduction in flatulence.	Study conducted in the early postoperative stage, with limited oral consumption of sugars and fats, which could affect SIBO and gastrointestinal symptoms.
Peinado Fabregat et al., 2022 [51]	19 patients with SIBO aged 1–21 years.	Probiotic therapy with *Lactobacillus rhamnosis* alone or in combination with antibiotics for 7–14 days.	Partial or complete symptom resolution in 81.2% of patients taking probiotics (alone or with antibiotics), compared to 67.7% in those treated with antibiotics alone.	Retrospective study; small sample size with wide age range (1–21 years); no breathing tests performed after treatment.
Ockeloen et al., 2012 [52]	10 pediatric patients (ages 1–18) with SIBO.	Daily probiotic supplementation for 8 weeks: 2 g of powder (1 × 10⁹ CFU/g) *Bifidobacterium* and *Lactobacillus* for ages 1–4, and one capsule (1 × 10⁹ CFU/g) *Bifidobacterium* and *Lactobacillus* for ages 5–18.	Reduced severity of abdominal complaints related to SIBO in 70% of children at 5 months and in 40% at 15 months after start of therapy. Authors suggested potential benefit of repeat probiotic cycles to maintain the effect.	Retrospective study with small sample size and wide age range (1–18 years).

**Table 2 microorganisms-13-00057-t002:** Probiotic treatment outcomes for SIBO in pregnant women with hypothyroidism.

Study	Hypothyroidism and Stage of Pregnancy	Number of Patients with SIBO Treated with Pre- and Probiotics	Probiotic Strains and Type of Prebiotic	Dosage and Duration of Administration	The Overall (and CH_4_) SIBO Negative Conversion Rate [%]	Limitations of the Study
Ouyang et al., 2024 [54]	Subclinical hypothyroidism; second trimester.	32	*Bifidobacterium infantis*, *Lactobacillus acidophilus*, *Enterococcus faecalis*, and *Bacillus cereus*; dietary supplement: inulin, ice sugar, microcrystalline cellulose, and oat fiber.	1.5 g, three times a day (probiotics); 5 g, three times a day (prebiotics); both for 21 days.	28.1 (63.9)	Small group size; no possibility of long-term follow-up at a specific stage of pregnancy.
Hao et al., 2022 [56]	Clinical hypothyroidism; second trimester.	112	*Bifidobacterium infantis*, *Lactobacillus acidophilus*, *Enterococcus faecalis*, and *Bacillus cereus*; dietary supplement: inulin, ice sugar, microcrystalline cellulose, and oat fiber.	1.5 g, three times a day (probiotics); 5 g, three times a day (prebiotics); both for 21 days.	53.6	No long-term assessment of SIBO possible.
Zhang et al., 2023 [55]	Hypothyroidism; <14 weeks.	28	*Bifidobacterium infantis*, *Lactobacillus acidophilus*, *Enterococcus faecalis*, and *Bacillus cereus*; no use of prebiotics.	1.5 g, three times a day for 21 days.	71.4 (90.7)	Small group size; no long-term SIBO assessment.

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
