# Peer review of "A Comprehensive Review of the Usefulness of Prebiotics, Probiotics, and Postbiotics in the Diagnosis and Treatment of Small Intestine Bacterial Overgrowth"

_microorganisms, 2025, doi:10.3390/microorganisms13010057_

Round 1

Reviewer 1 Report (Previous Reviewer 2)

Comments and Suggestions for Authors

Thanks Good job

Author Response

Dear Reviewer,

Thank you

Reviewer 2 Report (Previous Reviewer 1)

Comments and Suggestions for Authors

Dear Editor and Authors,

The manuscript has improved since its initial submission. However, I would like to make a few recommendations to clarify some points:

  1. Data Sources: Please state whether the database search was exhaustive and if grey literature or additional sources were included.

  2. Study Limitations: It would be helpful to critically evaluate the limitations of the included studies, such as small sample sizes or potential biases.

  3. Title Clarity: Consider whether including the study design in the title might enhance clarity and alignment with the manuscript's scope. For example, a title such as 'A Comprehensive Review of the Usefulness of Prebiotics, Probiotics, and Postbiotics in the Diagnosis and Treatment of Small Intestine Bacterial Overgrowth' could help readers better understand the nature of the study

Author Response

Reviewer 2

Thank you for your valuable comments.

Dear Editor and Authors,

The manuscript has improved since its initial submission. However, I would like to make a few recommendations to clarify some points:

  1. Data Sources: Please state whether the database search was exhaustive and if grey literature or additional sources were included.

We added that to the manuscript. Thank you

  1. Study Limitations: It would be helpful to critically evaluate the limitations of the included studies, such as small sample sizes or potential biases.

The chapter on study limitations was added.

  1. Title Clarity: Consider whether including the study design in the title might enhance clarity and alignment with the manuscript's scope. For example, a title such as 'A Comprehensive Review of the Usefulness of Prebiotics, Probiotics, and Postbiotics in the Diagnosis and Treatment of Small Intestine Bacterial Overgrowth' could help readers better understand the nature of the study

Thank you, the title was changed as you suggested.

This manuscript is a resubmission of an earlier submission. The following is a list of the peer review reports and author responses from that submission.

Round 1

Reviewer 1 Report

Comments and Suggestions for Authors

Dear Authors and Editor,

The manuscript entitled "Prebiotics, Probiotics, and Postbiotics in the Diagnosis and Treatment of Small Intestine Bacterial Overgrowth" aims to present current knowledge on prebiotics, probiotics, and postbiotics in the context of SIBO.

There are several minor and major issues that I would like the authors to address.

  1. Title:
    The authors do not specify the study design.

  2. Abstract:
    The abstract is not structured; it does not indicate the methodology used, the results obtained, or the conclusions. Additionally, keywords that are not found in the MeSH thesaurus have been included. It is recommended to replace these keywords with MeSH terms.

  3. Introduction:
    The authors adequately summarize the background of the study.

  4. Study Design:
    The type of study or design used is not explicitly mentioned. In a review, it is important to clearly state whether it is a systematic review, a narrative review, a rapid review, a state-of-the-art review, or a scoping review, and to detail the literature search methods to ensure thoroughness and rigor.

  5. Methodology:
    A clear methodology for the selection and analysis of the literature reviewed is not specified, which could call into question the completeness of the review and its reproducibility. It would be helpful to include the inclusion/exclusion criteria, databases consulted, and a study selection flowchart. All of this will depend on the type of review that the authors intended to conduct.

Author Response

Dear Reviewer

Thanks you very much for a such valuable review and helpfully comments. Based on you instruction we improved our manuscript. All changes are marks.

  1. Title:
    The authors do not specify the study design.

We changes the title of manuscript to “The usefulness of prebiotics, probiotics and postbiotics in the diagnosis and treatment of Small Intestine Bacterial Overgrowth.”

  1. Abstract:
    The abstract is not structured; it does not indicate the methodology used, the results obtained, or the conclusions. Additionally, keywords that are not found in the MeSH thesaurus have been included. It is recommended to replace these keywords with MeSH terms.

The new abstract are prepared, new keywords from MeSH are introduced.

  1. Introduction:
    The authors adequately summarize the background of the study.

Thank you

  1. Study Design:
    The type of study or design used is not explicitly mentioned. In a review, it is important to clearly state whether it is a systematic review, a narrative review, a rapid review, a state-of-the-art review, or a scoping review, and to detail the literature search methods to ensure thoroughness and rigor.

Thank you for this pertinent comment. The manuscript was written as comprehensive review. We added a new paragraph where we clarified the methodology.

  1. Methodology:
    A clear methodology for the selection and analysis of the literature reviewed is not specified, which could call into question the completeness of the review and its reproducibility. It would be helpful to include the inclusion/exclusion criteria, databases consulted, and a study selection flowchart. All of this will depend on the type of review that the authors intended to conduct.

As a above.

Reviewer 2 Report

Comments and Suggestions for Authors

The review submitted to “Microorganisms” MDPI journal, entitled “Prebiotics, probiotics and postbiotics in the diagnosis and treatment of Small Intestine Bacterial Overgrowth.” which study the current knowledge of prebiotics, probiotics and postbiotics in the context of SIBO. The following comments should be followed:

-         Please remove the point at the end of the title of the review

-         CH4 should make the number 4 subscript in whole the review.

-          Abstract is too little, so more details about the core investigation in the review should be added.

-         Line 17: correct beath to be breath.

-         Line 292: make this title non bold and italic.

-         References should be updated with relevant ones up to 2024.

-         The review is very interest that it is about a subject of great concern as a public health concern.

-         English writing is good.

-         The body of the review was written very well and with follow.

-         Conclusion is collaborative and highlight the importance of the review subject about the possible diagnosis and treatment of SIBO

Author Response

Dear Reviewer

Thanks you very much for a such valuable review and helpfully comments. Based on you instruction we improved our manuscript. All changes are marks.

The review submitted to “Microorganisms” MDPI journal, entitled “Prebiotics, probiotics and postbiotics in the diagnosis and treatment of Small Intestine Bacterial Overgrowth.” which study the current knowledge of prebiotics, probiotics and postbiotics in the context of SIBO. The following comments should be followed:

-         Please remove the point at the end of the title of the review

Done. Due to suggestions another reviewer, we change the title.

-         CH4 should make the number 4 subscript in whole the review.

Done.

-          Abstract is too little, so more details about the core investigation in the review should be added.

We change the abstract. Thank you for a suggestion.

-         Line 17: correct beath to be breath.

Done.

-         Line 292: make this title non bold and italic.

Done.

-         References should be updated with relevant ones up to 2024.

Dear Reviewer,  according to the methodology used, no more matching studies were found.

-         The review is very interest that it is about a subject of great concern as a public health concern.

Thank you.

-         English writing is good.

Thank you.

-         The body of the review was written very well and with follow.

Thank you.

-         Conclusion is collaborative and highlight the importance of the review subject about the possible diagnosis and treatment of SIBO

Thank you.